# Micro-Spatial Analysis of Maize Yield Gap Variability and Production Factors on Smallholder Farms

**Munialo Sussy [1],\*, Hall Ola [2], Francisca Archila Bustos Maria [2], Boke-Olén Niklas [3], Onyango M. Cecilia [1], Oluoch-Kosura Willis [4], Marstorp Håkan [5] and Göran Djurfeldt [2]**

[1] Department of Plant Science and Crop Protection, University of Nairobi, Nairobi P.O. Box 29053-00625, Kenya; cecilia.onyango@uonbi.ac.ke

[2] Department of Human Geography, Lund University, 22100 Lund, Sweden; ola.hall@keg.lu.se (H.O.); maria.archila@keg.lu.se (F.A.B.M.); goran.djurfeldt@keg.lu.se (G.D.)

[3] Centre for Environmental and Climate research (CEC), Lund University, 22100 Lund, Sweden; niklas.boke_olen@cec.lu.se

[4] Department of Agricultural Economics, University of Nairobi, Nairobi P.O. Box 29053-00625, Kenya; willis.kosura@gmail.com

[5] Department of Soil and Environment, Swedish University of Agricultural Sciences, S-75007 Uppsala, Sweden; hakan.marstorp@slu.se

\* Correspondence: munialos@yahoo.co.uk; Tel.: +254-7105-81953

**Abstract:** Site-specific land management practice taking into account variability in maize yield gaps (the difference between yields in the 90th percentiles and other yields on smallholder farmers' fields) could improve resource use efficiency and enhance yields. However, the applicability of the practice is constrained by inability to identify patterns of resource utilization to target application of resources to more responsive fields. The study focus was to map yield gaps on smallholder fields based on identified spatial arrangements differentiated by distance from the smallholder homestead and understand field-specific utilization of production factors. This was aimed at understanding field variability based on yield gap mapping patterns in order to enhance resource use efficiency on smallholder farms. The study was done in two villages, Mukuyu and Shikomoli, with high and low agroecology regarding soil fertility in Western Kenya. Identification of spatial arrangements at 40 m, 80 m, 150 m and 300 m distance from the homestead on smallholder farms for 70 households was done. The spatial arrangements were then classified into near house, mid farm and far farm basing on distance from the homestead. For each spatial arrangement, Landsat sensors acquired via satellite imagery were processed to generate yield gap maps. The focal statistics analysis method using the neighborhoods function was then applied to generate yield gap maps at the different spatial arrangements identified above. Socio-economic, management and biophysical factors were determined, and maize yields estimated at each spatial arrangement. Heterogeneous patterns of high, average and low yield gaps were found in spatial arrangements at the 40 m and 80 m distances. Nearly homogenous patterns tending towards median yield gap values were found in spatial arrangements that were located at the 150 m and 300 m. These patterns correspondingly depicted field-specific utilization of management and socio-economic factors. Field level management practices and socio-economic factors such as application of inorganic fertilizer, high frequency of weed control, early land preparation, high proportion of hired and family labor use and allocation of large land sizes were utilized in spatial arrangements at 150 and 300 m distances. High proportions of organic fertilizer and family labor use were utilized in spatial arrangements at 40 and 80 m distances. The findings thus show that smallholder farmers preferentially manage the application of socio-economic and management factors in spatial arrangements further from the homestead compared to fields closer to the homestead which could be exacerbating maize yield gaps. Delineating management zones based on yield gap patterns at the different spatial arrangements on smallholder farms could contribute to site-specific land management and enhance yields. Investigating the value smallholder

farmers attach to each spatial arrangement is further needed to enhance the spatial understanding of yield gap variation on smallholder farms.

**Keywords:** spatial arrangements; heterogeneous farms; yield gap patterns; site-specific; land management; unequal resource

## 1. Introduction

Smallholder farmers contribute approximately 75% of agricultural productivity and employment in many parts across the world [1]. However, these farmers live on farms that are less than 2 hectares which are highly heterogeneous with regard to soil quality, productive assets and technology [2]. These diversities contribute to significantly higher maize yield gaps (the difference between yields in the 90th percentiles and other yields on smallholder farmers' fields) greater than 50% which continue to persist causing food insecurity [3]. Understanding yield gap variability and the causes can enhance site-specific land management and improve yields [4]. However, there is limited understanding of the causes of yield gaps at a micro-level. This is because studies on analysis of yield gaps at a local level have used methods such as surveys and field experimentation to understand factors limiting crop yields [5,6]. These methods have spatial data limitation where only few randomly sampled units are used and fail to provide a comprehensive understanding of yield gaps at micro-level considering diversity which exists even within fields and plots [7].

Remote sensing has the ability to overcome spatial data limitation and can complement surveys or field experimentations in understanding yield gap variability [7]. Remote sensing has been successfully bused to generate yield maps and enabled application of site-specific management on homogenous farms [8,9]. A few studies have reported using remote sensing technology to map yield and yield gaps on smallholder farms [10,11]. However, diversity in topography, land sizes and management practices are still challenges hampering utilization of remote sensing on smallholder farms as far as the spatial understanding of yield gaps and their causes is concerned [10]. Identifying patterns with nearly similar yields can help in the creation of management zones that could be managed uniformly, thus promoting site-specific land management [12].

Site-specific land management where inputs such as fertilizer and herbicides are applied within fields can reduce waste, maintain environmental quality and sustain crop production [4]. Site-specific land management premises on spatial dependence which assumes that near things are closely related than distant things [13]. On the other hand, smallholder farming systems are characterized by a unique spatial arrangement where fields are located at proximity to the homestead, at the middle and at the further ends of the farm [14]. These spatial arrangements which are differentiated by distance from the homestead affect utilization of management and socio-economic factors and in occurrence of soil factors which affect yields [14]. Mapping patterns of yield gaps at the different spatial arrangements can aid in investigating field-specific management, in soil as well socio-economic factors and resource utilization patterns. This requires investigation using high resolution imagery and spatial analysis methods that can provide information at finer details.

High resolution multispectral imagery such as Landsat sensors acquired via satellite imagery are becoming plausible for investigating maize yield gaps on heterogeneous farming systems [15]. Focal statistics analysis is one of the approaches that has been utilized to show fine detailed information in health studies [16]. Focal statistics analysis performs a neighborhood operation at different distances resulting in output raster maps where the value for each output cell is a function of the values of all the input cells that are in a specified neighborhood around that location [13]. This function can thus help cluster patterns of yield gaps at the different spatial arrangements with respect to distance on smallholder farms which will provide a wide range of information and aid in field management

decision making. Nonetheless, studies mapping yield and yield gaps at a local level are yet to consider the spatial arrangements found on smallholder farms.

The purpose of the study was to improve the spatial understanding of yield gaps and their causes using spatial arrangements found on heterogeneous farming systems complemented with survey data as scope for promoting site-specific land management and enhancing yields. The use of spatial arrangements to map yield gaps on smallholder farms is a unique approach which contributes to the existing knowledge on use of remote sensing in mapping of yield gaps at micro-level. The study answered the following two research questions. How do spatial arrangement on smallholder farms affect the maize yield gaps? Are management, socio-economic and biophysical factors inclined towards certain spatial arrangements?

## 2. Materials and Methods

The study used a unique approach where yield gap maps were created at different spatial arrangements and correspondingly identified field management, socio-economic and biophysical factors at each arrangement. The study was important to provide field-specific soil and crop management measures. The structure of the materials and methods section was as follows:

- Description of the study sites;
- Collection and analysis of field data;
- Collection and analysis of remote sensing data.

### 2.1. Description of the Study Sites

The study was conducted in two sites, Mukuyu and Shikomoli of Kakamega and Vihiga counties, respectively (Figure 1). The two villages were drawn from the Intensification of the Africa Project (Afrint). The initial selection and sampling of the sites is described by [17]. The sites have agricultural intensification potential yet are dynamic in terms of agro-ecology, population density and market accessibility. Mukuyu has high agro-ecological potential, however market accessibility is poor, while Shikomoli has low agro-ecological potential with fairly good market access [18].

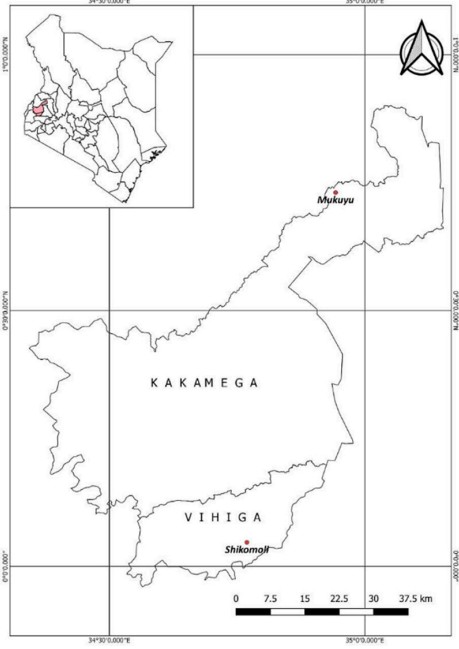

**Figure 1.** Map showing the study regions in Kenya (Mukuyu: Latitude 0°43′50″ N, Longitude 34°56′34″ E. Shikomoli: Latitude 0°02′49″ N, Longitude 34°46′09″ E).

### 2.2. Mukuyu Village

Mukuyu village is located in Lugari Sub County in Kakamega County (0°43'50" N, 34°56'34" E) with an altitude of 1600 m above sea level. The village is classified into Upper Midland Zone 4. The annual rainfall ranges between 1000 mm and 1600 mm with a mean of 1450 mm per annum. The rain is bimodal with long rains occurring between March and July and short rains occurring between September and November. The rest of the months are partially dry. During these dry months most of the land is left fallow; however, in some places beans are grown [19]. The daily temperature varies between 14 and 26 °C with a mean of 20 °C. The soils range from Ferralsols, which are dark red, to Acrisols, which are dark reddish brown [20]. These soils are suited for agricultural use [21].

Mukuyu village has a low population density (373 people/km$^2$), fairly large farm sizes (average 3 acres) and good general soil fertility [22,23]. The main crops grown include maize, beans, sweet potatoes, cassava, sorghum and millet as food crops and coffee, sugar cane, sunflower and a variety of fruits for income. Maize is the staple food crops and farmers prefer to grow long season maize varieties of 6–8 months due to high yielding characteristic [19]. Crop production is mainly rain-fed although some irrigation is in practice [18]. Other farming activities undertaken include poultry keeping, bee keeping, fish farming and dairy farming [24]. Manual labor with hand tools is used while in some incidences farmers use tractors, especially for land preparation before planting. Inorganic and organic fertilizers are used for fertility improvement [24].

### 2.3. Shikomoli Village

Shikomoli village is located in Vihiga County (0°02'49" N, 34°46'09" E) at an altitude of 1400 m above sea level. The village is located on the western region of Kenya, in the Lake Victoria Basin. The equator cuts across the southern part of the village. Generally, the village has undulating hills and valleys with streams flowing from northeast to southwest and draining into Lake Victoria. The amount of rainfall received in the county ranges between 1600 mm to 2000 mm with a mean of 1700 mm per annum. The rainfall is bimodal having a long rain season that occurs between the months of April and June and a short rain season that occurs between September and November. The daily temperature ranges between 14 and 32 °C with mean temperature of 23 °C. The village has relatively poor agricultural potential (the area is rocky and hilly) having poorly developed soils. The main soil types are Nitisols, Cambisols and Acrisols [20]. The soils are constraints for agricultural production due to their shallow depths [21].

Shikomoli village has a very high population density (2100 people/km$^2$) [22,23]. Women make up the largest size of the population [23]. The average farm size is 1 acre with a few farmers having farm sizes of approximately 4 acres [22]. The main food crops produced are maize intercropped with beans which are produced both during the long and short rain season. Although maize is the main staple crop in the village, the yields per acre are extremely low, approximately less than 1 t/ha. Other crops include sweet potatoes, sorghum, finger millet and groundnuts. Cash crops grown include tea, coffee and horticultural crops. Other agricultural activities include dairy cattle, goats, sheep and poultry [25]. The crop production system in this village is largely subsistence-based and not market-oriented [18]. Production is based on the use of traditional cultivars and even when improved cultivars are used, management practices are similar to those used in the production of the local cultivars. The farmers use labor intensive techniques and there is little to no use of fertilizers for soil fertility improvement, nor do farmers use chemicals for pest and disease control [26]. The farmers practice rain-fed agriculture. The sloping terrain hampers effective farming as soils are subject to higher rates of water runoff and soil erosion [26].

### 2.4. Collection and Analysis of Field Data

Maize fields were identified and georeferenced with GPS from 70 households in Mukuyu and Shikomoli, respectively. The total number of maize fields was 170. After identification a 4 m by 4 m

area was marked at the center of each identified maize field and acted as the study plot on which maize yields as well as biophysical, management and socio-economic factors were measured.

The study identified three spatial arrangements; Near House (NH), Mid Farm (M) and Far Farm (FF). The NH pattern was a piece of land located close to the main household, M pattern was a piece of land located next to the NH but at a far distance from the main household and FF pattern was a piece of land located next to the M but at further distance from the main household as shown in Figure 2. The distance from the homestead to the maize fields located at the near homestead, middle farm and far end was measured and recorded. The farmers helped identify the plots that belonged near house, at mid farm and far farm.

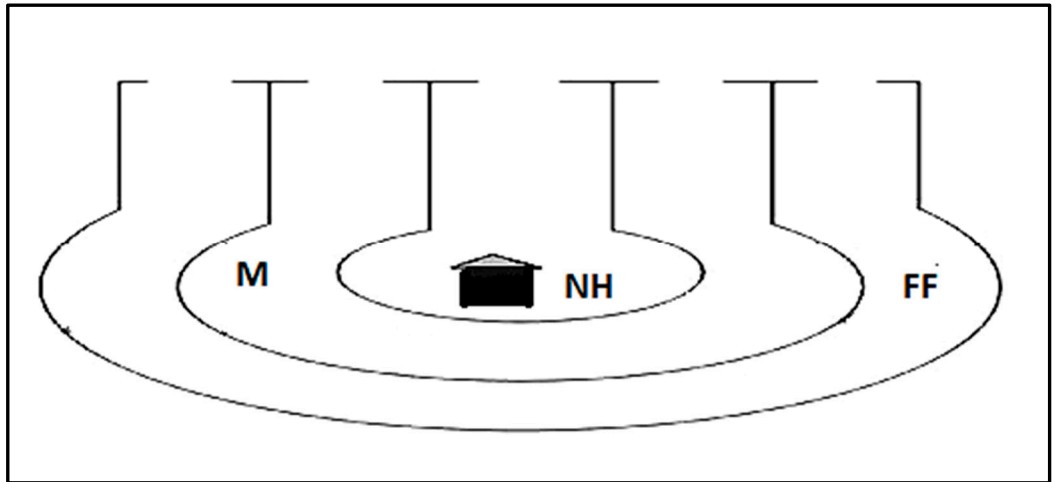

**Figure 2.** Spatial arrangements on smallholder farming systems. Legend: NH—Near House pattern, M—Mid Farm pattern, FF—Far Farm pattern. These are spatial arrangements that were identified on smallholder farms. Source: Authors own development.

Socio-economic, management and biophysical factors at each spatial arrangement were also collected and are described in Table 1. Socio-economic factors included land size, labor use, gender and credit facility. Management factors were inorganic and organic fertilizer use, maize variety, land preparation and weed control. Both socio-economic and management data were collected using a household survey for each spatial arrangement. Biophysical factors were measured from the 4 m by 4 m plot delineated on each spatial arrangement and included maize density, soil plant analysis development (SPAD) values (chlorophyll content), weed pressure (weed cover and height), maize height, slope, erosion status and soil properties (phosphorus (P), nitrogen (N) and boron (B)). Soil properties were sampled, processed and analyzed for soil nutrients at Crops and Nutrition Laboratory Services in Nairobi. Soil properties together with slope were determined at the start of the maize growing period. While maize density, SPAD values (chlorophyll content), weed pressure (weed cover and height), maize height and erosion status were measured at the ear initiation and at silking and tasseling, corresponding to maize development stages 1 and 3, respectively, according to [27].

**Table 1.** Socio-economic, biophysical and management factors collected.

| Variables | Description |
|---|---|
| Total land size (TTLs) | Size of the cultivable land in acres (whether inherited, leased or purchased) owned by the farmer. |
| Labor use | Family and hired labor used for all operations related to maize cultivation (man hour ha$^{-1}$); categorized as 1—Family, 2—Hired. |
| Gender of farm operator | The state of the farm operator being male (=1), or female (=2). |
| Credit facility | Credit acquisition for use on farm activities; Yes = 1, Otherwise = 0. |
| Inorganic | Quantity and frequency of inorganic fertilizer use; Yes = 1, Otherwise = 0 |
| Organic | Quantity of organic fertilizer use; Yes = 1, Otherwise = 0 |
| Land preparation | Time of preparing land for planting maize. 1—Before harvesting of the previous crop, 2—Immediately after harvesting, 3—2 Months before onset of rains, 4—1 month before onset of rains, 5—at the onset of rain, 6—1 week after the onset of rain, 7—2 weeks after onset of rains. |
| Maize variety | The duration of maize growth from planting to maturity; 1—long duration, 2—medium duration, 3—short duration |
| Frequency of weed control | Number of times weed control is done on the farm |
| Maize density | Number of maize plants per hectare. Determined through counting in the 4 m by 4 m plot quantified per hectare |
| Maize height | Measured on 10 randomly chosen plants in the 4 m by 4 m plot |
| Weed cover | Measured using a Likert scale according to [28]. |
| Weed height | Measured on 10 randomly chosen weeds in the 4 m by 4 m plot |
| SPAD values (chlorophyll content) | Measured using a SPAD 502 chlorophyll meter (Minolta Camera Co., Osaka, Japan) by taking readings of the youngest fully developed leaf from 15 randomly selected plants per study plot, at approximately 25% from the leaf tip and leaf base. |
| Soil properties | Soil nutrients; nitrogen (N), boron (B), phosphorus (P) determined by methods described by [29]. |
| Slope | Measured using a Likert scale 1–3 where 1—steep, 2—gentle, 3—flat. Erosion values of 0—none, 1—slight, 2—moderate, 3—severe, according to [30]. |
| Erosion status | Measured using a Likert scale 0–3 where 0—none, 1—slight, 2—moderate, 3—severe, according to [30]. |

Legend: (SPAD—Soil plant analysis development).

Maize yield was estimated on a dry matter basis using the method described by [31]. Maize in the study plots (4 m by 4 m plot earlier identified) was harvested and the grain shelled, cleaned, weighed and recorded in kg. A subsample of approximately 200 g was oven dried at 75 °C for 24 h and weighed. The subsample was used to determine moisture content and to calculate the yield as kg dry matter for the harvested area. The estimated yields in the study plot was then extrapolated in tonnes per hectare. Maize yields were then classified into different percentiles; 90th, 75th, 50th and 25th using the formula:

$$K^{th} = L\left[\frac{(P - cfb)}{f}\right] \tag{1}$$

where

$K^{th}$ = the percentile to be calculated.

$L$ = the lower limit of the critical value within which the percentile will occur.

$P$ = (K/100) (*n*) where *K* is the percentile and *n* is the number of values in the distribution. *P* is the critical interval where the percentile (*K*) will occur.

*cfb* = the cumulative frequency of all intervals below the critical value but not including the critical value.

*F* = the frequency in the critical interval.

*U* = the upper limit of the critical value that will not be included in the critical interval.

### 2.5. Collection and Analysis of Remote Sensing Data

The collection and analysis of remote sensing data involved acquisition of satellite and Landsat 8 images, image preparation, processing of the images to yield maps, and validation of yields. The process is described by [32]. Two satellites images with four bands, Blue–Green–Red–NIR, obtained from TerraNor in Roa Norway, were acquired on 19 June 2016 by GeoEye 1 for Mukuyu and Shikomoli. Two Cloud-free Landsat 8 Collection 1 Level-2 on-demand surface reflection data were obtained through Earth Explorer. For Shikomoli, the image was taken on 30 June 2016 while the image for Mukuyu was taken on 14 June 2016. The images were projected to UTM projection (Zone 36 N) using the WGS84 datum (United States National Geospatial-Intelligence Agency, Virginia, US). Clouds were removed from the Landsat 8 images using the image classification procedure in ArcGIS. The procedure identified a training sample set which was used to classify clouds and no clouds images. The clouds images were then used to mask clouds from the original fine resolution satellite image. Radiometric correction to surface reflectance was then done using the method described by [11]. The histogram matching process was undertaken for the four bands (Red–Green–Blue–NIR) in ERDAS Imagine software and this resulted in a composite surface reflectance image with four bands.

The green chlorophyll vegetation index (GCVI) was the vegetation index that was used to map yields, and it was calculated according to [11]. The Agricultural Production Systems sIMulator (APSIM) was then used to generate pseudo observations for yield [11]. Yields were then estimated following the Scalable satellite-based Crop Yield Mapper (SCYM) methodology and yield maps drawn [33].

The outputs were then isolated to only maize fields as shown in Figures A1 and A2 (see Appendix A). This involved creating a land cover classification mask using random forests classification, following [11]. This was done in R following a tutorial by Ali Santacruz [34]. The random forest classifier was trained using the known locations of maize fields, taken from the yield gap data (YGP), as well as visual inspection of the fine resolution imagery to identify trees and urban or non-natural areas. The classified image was then used to mask out all pixels that were classified as non-maize from the estimated yield image.

Validation of the final maize yield maps was done by comparing the estimated yields to the observed yields, using adjusted $R^2$ to quantify the agreement between the two. The observed yields in kg/ha were calculated for each of the 4 × 4 m quadrants from the YGD by dividing the yield in kg by 0.0016 ha (the assumed size of each quadrant). The quadrants' yields were assumed to be representative of the yields for the entire plot. The estimated yields were calculated as the average yield for all of the pixels located within each plot. Outliers were removed from both the estimated and observed yields in order to ensure that both datasets met the normality assumption of linear regression analysis.

### 2.6. Yield Gap Pattern Mapping

This process involved creating a constant yield map and creating yield gap maps at the different spatial patterns. The 90th percentile yield values identified earlier for each site were used to create a constant yield map using the raster creation tool in the spatial analyst of ArcGIS 10.1 from ESRI, Redlands, Calfornia, USA [35]. The maximum yields, i.e., the 90th percentile, provides an estimate of the genotype and environment interaction representative of the production system of smallholder farms, and hence is a better estimate of yield gaps compared to optimal yield data from experimental stations. A yield gap map for each site was then created by comparing the yield map earlier generated

versus the constant yield map using the map algebra function. The focal statistics in the neighborhood function was then used to generate yield gap maps at the different spatial arrangements (near house, mid farm and far farm) where for each, the average distance from the homestead was used as the input value for height and width fields in the neighborhood settings function. The focal statistics method is described by [36]. For each spatial arrangement, variability in yield gaps was computed using standard deviation, mean, maximum value and minimum value using the focal statistics function.

## 3. Results

### 3.1. Mapping Maize Yields in Mukuyu and Shikomoli

Figures 3 and 4 show variability in yields, highlighting several plots. The estimated pixel-level yield in Shikomoli was in the range of 0.08 t/ha and 4.9 t/ha, with an average of 2.2 t/ha and a median of 2.1 t/ha. In Mukuyu, the estimated pixel-level yield ranged from 1.1 t/ha to 5.5 t/ha, with an average of 2.6 t/ha and a median of 2.5 t/ha.

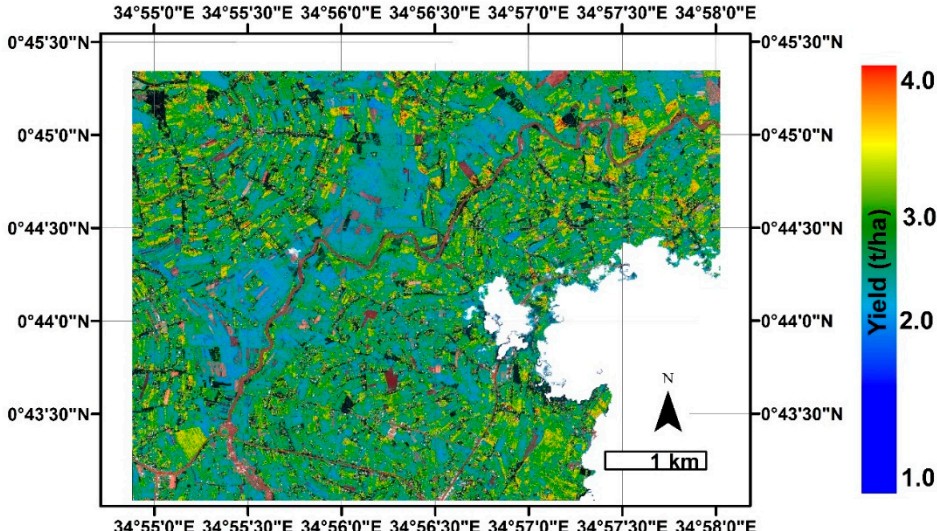

**Figure 3.** Yield map showing distribution patterns of high and low yields in Mukuyu. Source [32].

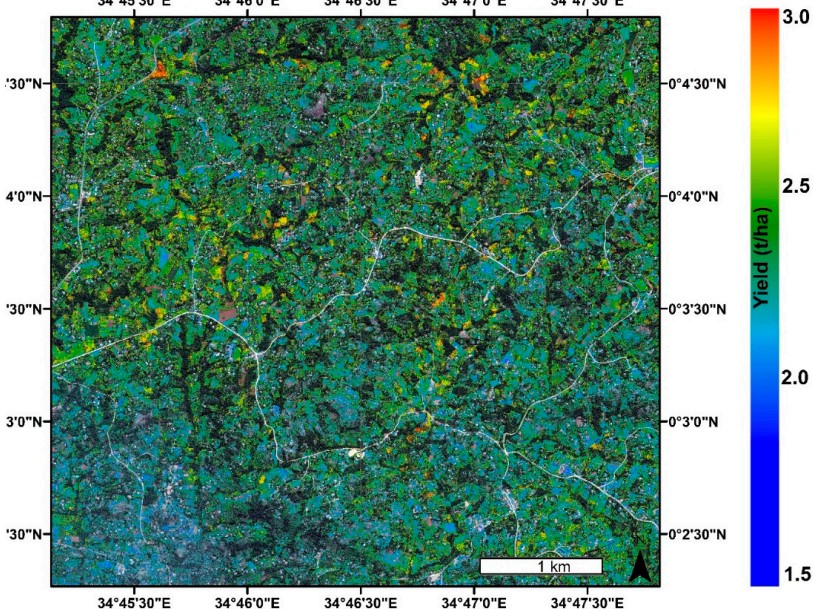

**Figure 4.** Yield map showing distribution patterns of high and low yields in Shikomoli. Source: [32].

### 3.2. Mapping Maize Yield Gaps in Mukuyu and Shikomoli

The 90th percentile yield values which were used to create constant yield maps were 5.1 and 4.8 t/ha for Mukuyu and Shikomoli. Figures 5 and 6 show yield gap mapping patterns derived from comparing the yield map (Figures 3 and 4) versus a constant yield maps. The yield gap map showed different patterns of low and large yield gaps. The minimum and maximum yield gap values were −1.0 and 3.3 t/ha for Mukuyu and −0.9 and 3.9 for Shikomoli. From the actual estimated yields some fields had yield values beyond the 90th percentile values as described in Section 3.1, hence the negative yield gap values.

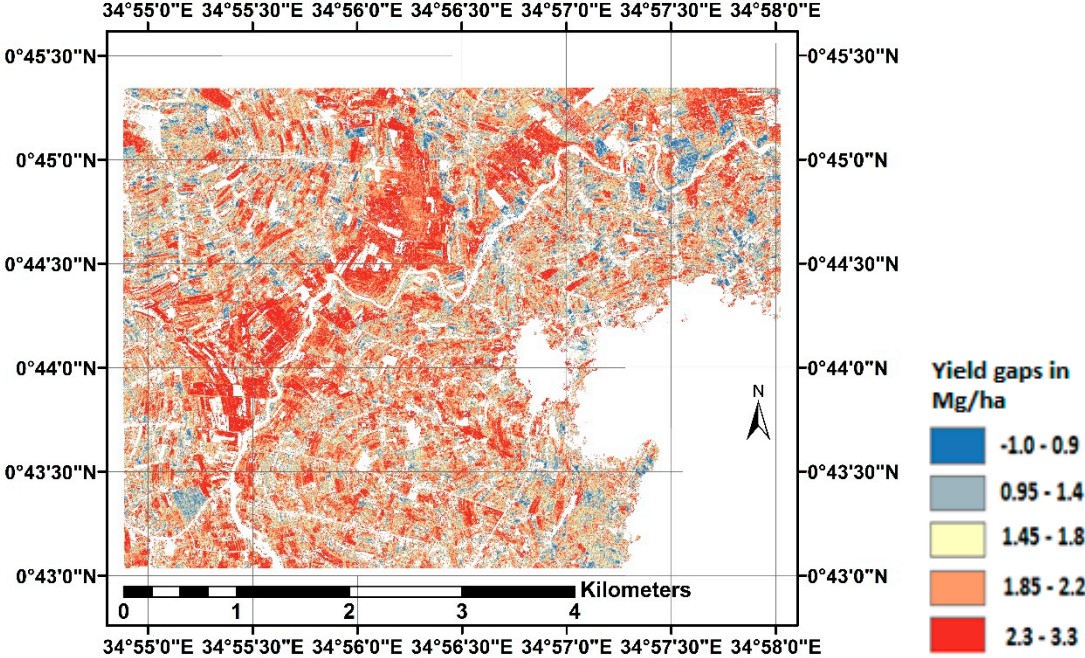

**Figure 5.** Yield gap map showing low, average and high yield gaps distribution patterns in Mukuyu.

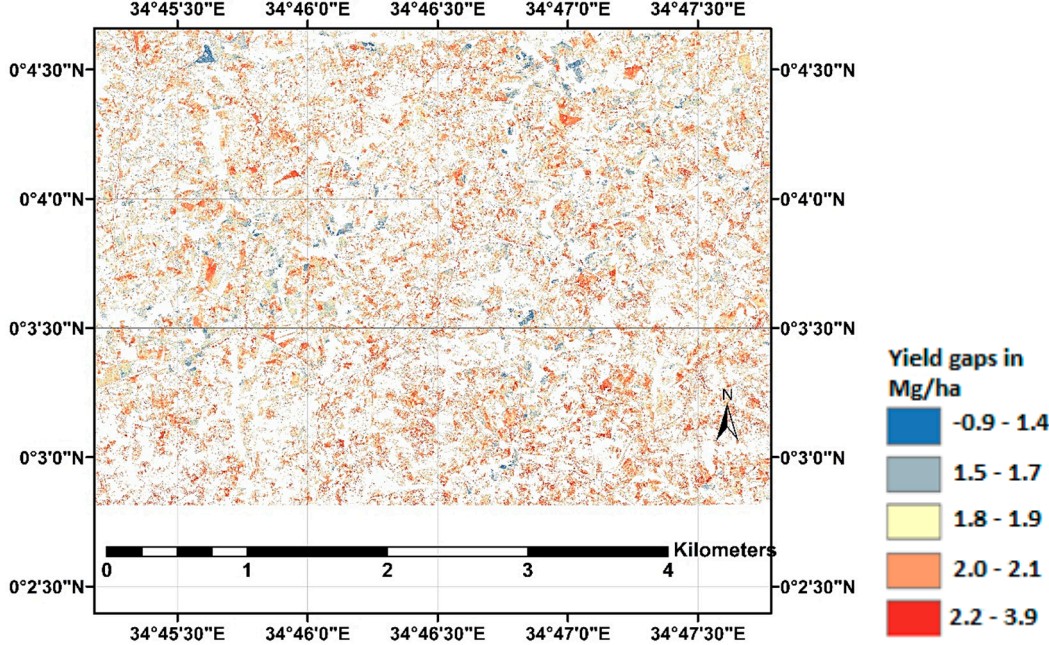

**Figure 6.** Yield gap map showing low, average and high yield gaps distribution patterns in Shikomoli.

### 3.3. Yield Gap Maps at Different Neighborhoods in Mukuyu and Shikomoli

Yield gaps were generated at different spatial arrangements with respect to distance from the homestead. The average distances of the three types of spatial arrangements on smallholder farms are shown in Table 2. High, average and low yield gaps were identified in spatial arrangements closer to the homestead (40 m by 40 m and 80 m by 80 m) (Figures 7 and 8). As distance increased, the high and low yield gap patterns stretched (150 m by 150 m and 300 m by 300 m) patterns towards average values (Figures 7 and 8).

**Table 2.** Spatial arrangements on smallholder farms.

| Mukuyu | Shikomoli | Plot Location |
|---|---|---|
| 40 m by 40 m | 40 m by 40 m | Near house |
| 80 m by 80 m | 80 m by 80 m | Mid farm |
| 150 m by 150 m | 150 m by 150 m | Far farm |
| 300 m by 300 m | 300 m by 300 m | Far farm |

Plot location is the distance from the smallholder homestead.

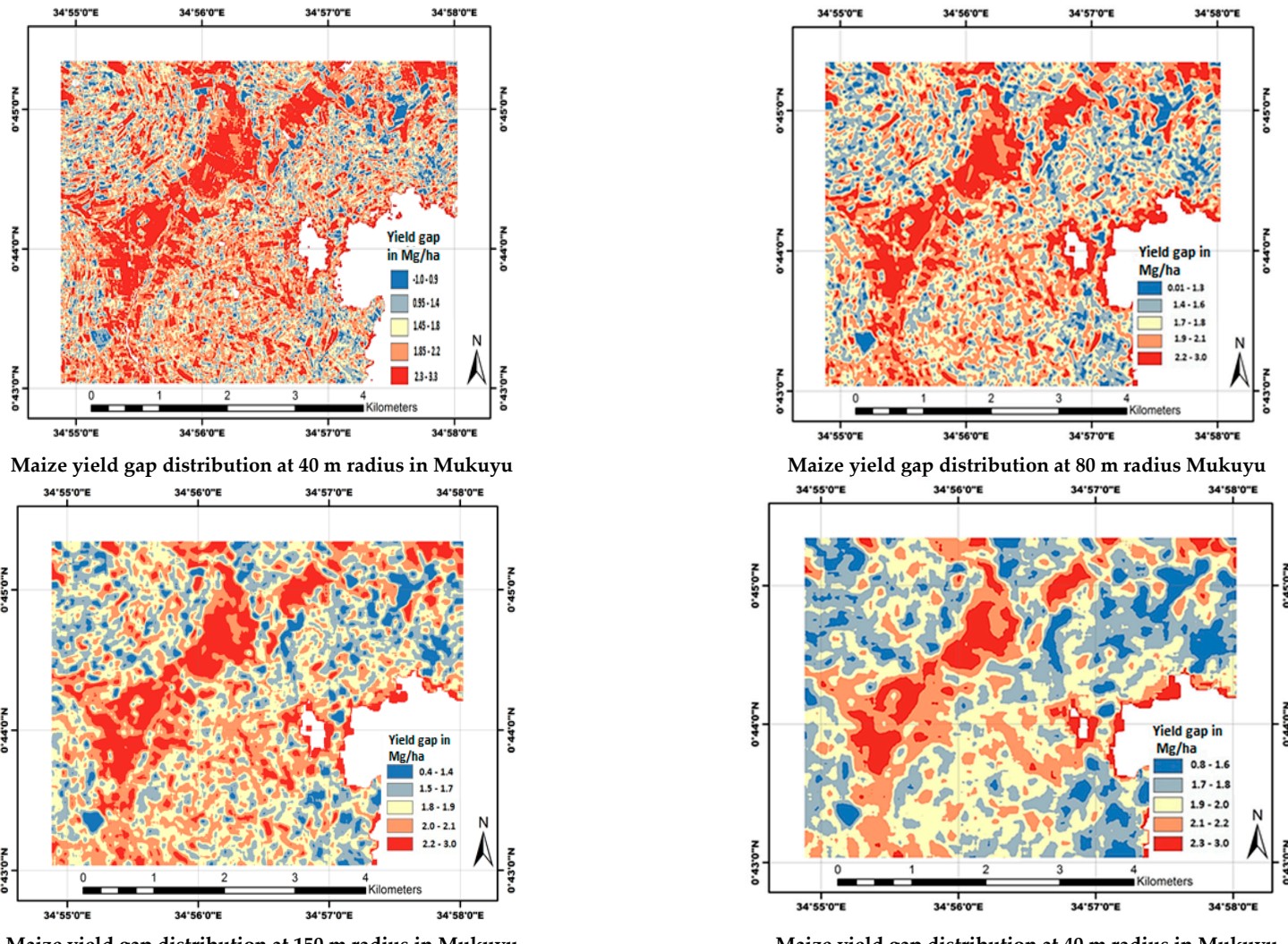

**Figure 7.** Yield gap mapping patterns at different spatial arrangements in Mukuyu. The blue and red regions represent patterns of low and high yield gaps which widen as distance increases.

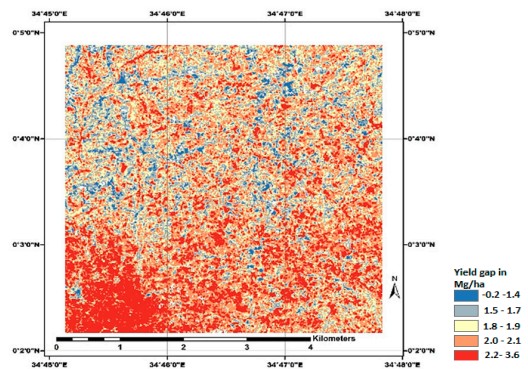

**Maize yield gap distribution at 40 m radius in Shikomoli.**

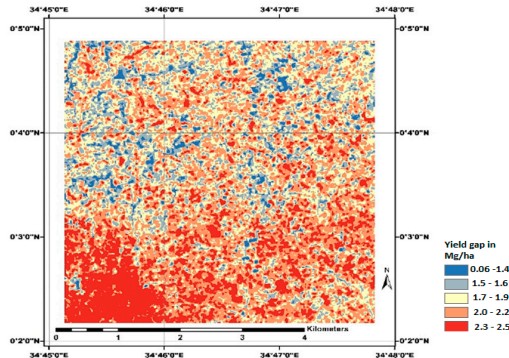

**Maize yield gap distribution at 80 m radius in Shikomoli.**

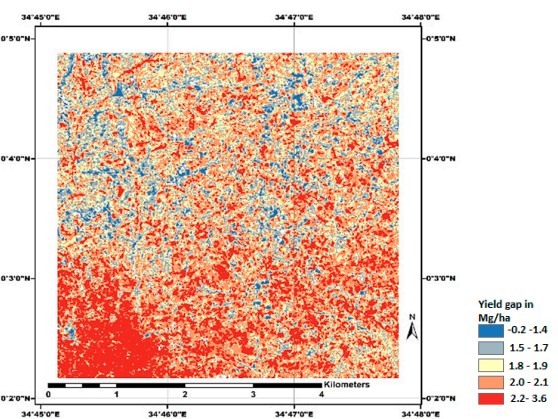

**Maize yield gap distribution at 150 m radius in Shikomoli.**

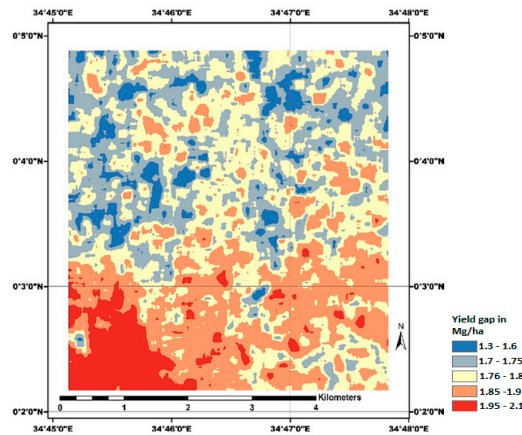

**Maize yield gap distribution at 300 m radius in Shikomoli.**

**Figure 8.** Yield gap mapping patterns at different spatial arrangements in Shikomoli. The purple and green regions represent patterns of low and high yield gaps.

### 3.4. The Maximum, Minimum and Mean Values and Variance at Different Spatial Arrangements

The maximum and minimum values of yield gaps in Mukuyu and Shikomoli for the different neighborhoods are shown in Table 3 The minimum values of yield gaps increased while the maximum values decreased with increasing distance from the homestead (Table 3). The mean values decreased as distance increased from the homestead.

The variance of maize yield gaps for plots that were close to the homestead was high and it decreased with increasing distance from the homestead for Mukuyu. In Shikomoli, there was a downward decrease and then an upward shift in variance after the 150 m distance. This indicates that spatial arrangements closer to the homestead exhibited heterogeneous patterns of low and high yield gaps, while arrangements further from the homestead had nearly homogeneous patterns of either high or low or average yield gaps. The plot variance was high in Mukuyu compared to Shikomoli (Figure 9).

**Table 3.** The minimum (min), mean and maximum (max) values of yield gaps in Mukuyu and Shikomoli at different spatial arrangements.

| Neighborhoods | Mukuyu | | | Shikomoli | | |
|---|---|---|---|---|---|---|
| | Max Values | Min Values | Mean Values | Max Values | Min Values | Mean Values |
| 40 m by 40 m | 3.3 | −1.0 | 1.9 | 3.6 | −0.2 | 1.85 |
| 80 m by 80 m | 3.0 | −0.1 | 1.89 | 2.4 | 0.06 | 1.84 |
| 150 m by 150 m | 3.0 | 0.4 | 1.88 | 2.2 | 1.0 | 1.84 |
| 300 m by 300 m | 3.0 | 0.8 | 1.87 | 2.2 | 1.3 | 1.83 |

Mean values of maize yield gaps at different spatial arrangements for Mukuyu and Shikomoli. The values are in t/ha. *t*-test statistics at 0.95 show mean values for maize yield gaps significantly different ($p = 0.001$) between the near house and mid farm, mid farm and far farm, and near house and far farm plots.

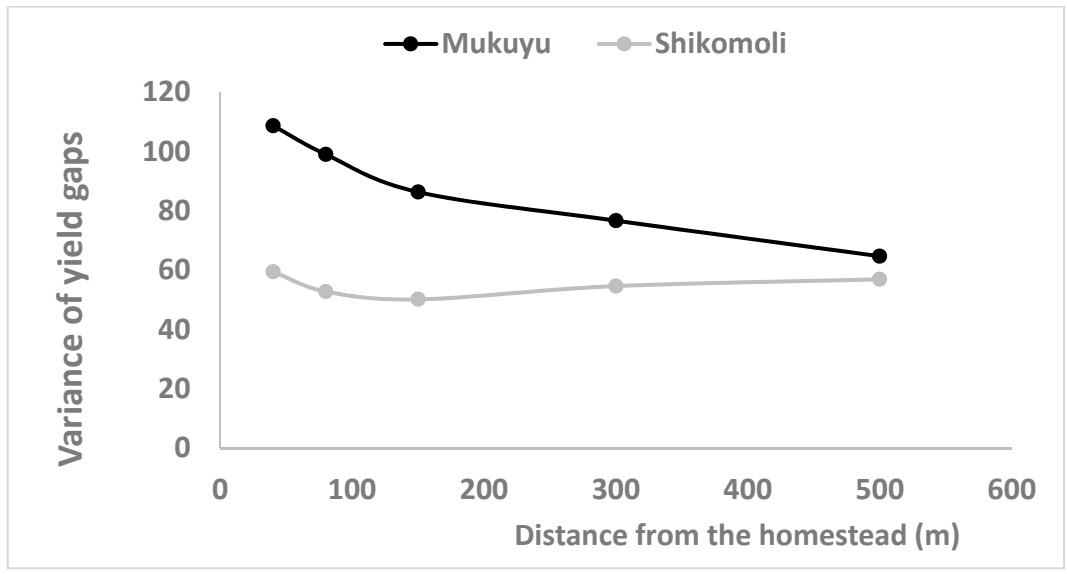

**Figure 9.** The variance in yield gaps at different spatial arrangements represented by distance from the homestead. High variance indicates heterogeneous patterns of low and high yield gaps. Low variance indicates nearly homogenous patterns of yield gaps.

### 3.5. Management, Biophysical and Socio-Economic Factors at Spatial Arrangements

There was variation in management practices and biophysical factors at different spatial arrangements (Figure 10). In Mukuyu, farmers preferred to plant medium and long variety crops at the mid and far farm, while the short variety was grown at the near house plots. In Shikomoli, the short and medium maize varieties were grown at the mid and far farm spatial arrangements. High amount of fertilizer was applied on the mid farm and near house plots in Mukuyu and Shikomoli

respectively. In both Mukuyu and Shikomoli, near house plots had high maize densities which decreased with increasing distance. In Shikomoli, the mid farm plots had high maize densities. In Shikomoli, the proportion of organic manure application was high for the near house plots compared to mid and far farm plots. The intensities of weed control increased and decreased for Mukuyu and Shikomoli respectively as distance increased from the homestead. In Mukuyu, there was delay in land preparation time for the near house plots compared to mid and far farm, while in Shikomoli, early land preparation was done on the near house plots. The plots in Shikomoli showed high erosion signs which were especially visible for the near and far farm plots that had a steep slope (Figure 10). Phosphorus, nitrogen and SPAD (Chlorophyll content) values increased with increasing distance from the homestead in Mukuyu (Table A1). Weed coverage was reduced with increasing distance from the homestead in Mukuyu. In Shikomoli, phosphorus, boron, SPAD (Chlorophyll content) values decreased while weed coverage increased with increasing distance from the homestead (Table A2).

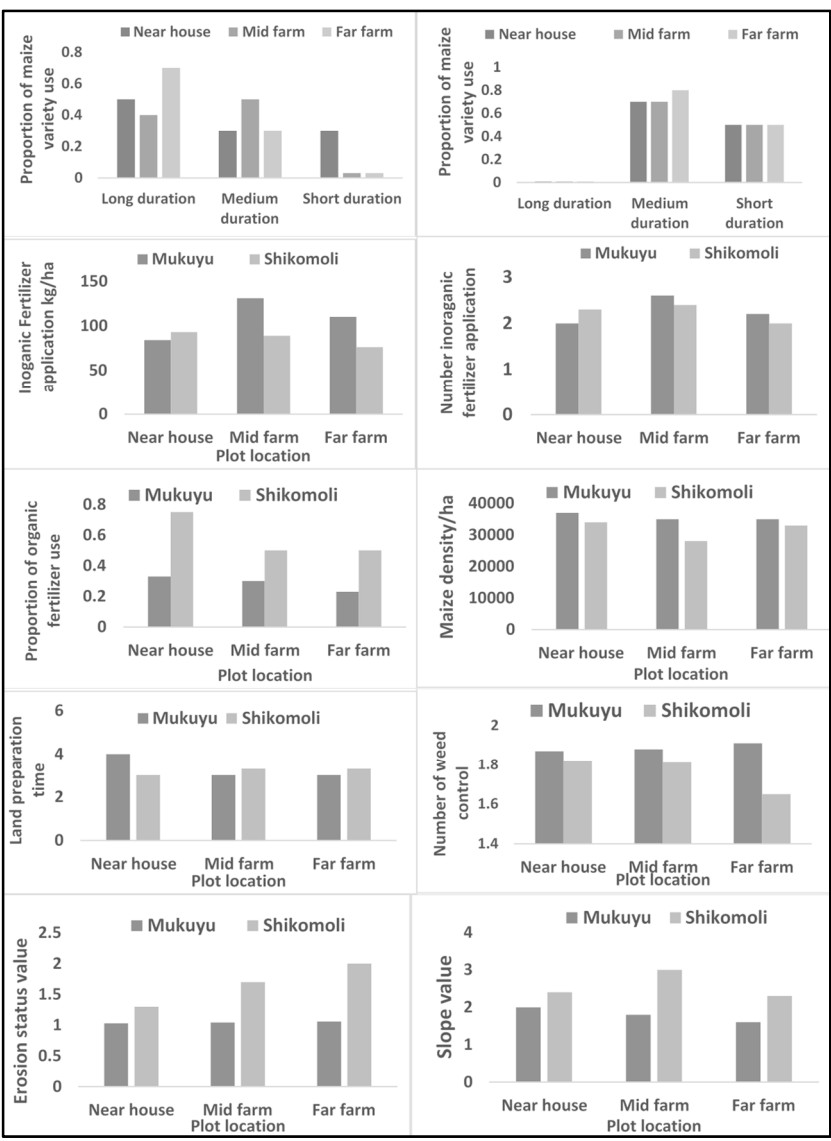

**Figure 10.** Management practices at different spatial arrangements. Land preparation (1—before harvesting of previous crop, 2—two months before onset of rain, 3—one month before onset of rain, 4—at onset of rain), slope status (1—steep, 2—gentle, 3—flat), erosion value— (0—none, 1—slight, 2—moderate, 3—severe). *t*-test statistics at 0.95 shows significant differences ($p = 0.001$) in occurrence in management factors between sites and spatial arrangements.

Figure 11 shows the socio-economic factors at different spatial arrangements (plot location). Land allocation to the near house, mid and far farm plots followed the same patterns for Mukuyu and Shikomoli. There was high allocation of land for the mid and far farm plots in Mukuyu compared to Shikomoli. In both sites farmers were likely to acquire credit facilities for the mid farm plots than for the near house and far farm plots. In Mukuyu, there was higher utilization of hired labor at the mid and far farm plots compared to the near house. In Shikomoli, more hired labor was used at the near house plots compared to mid and far farm.

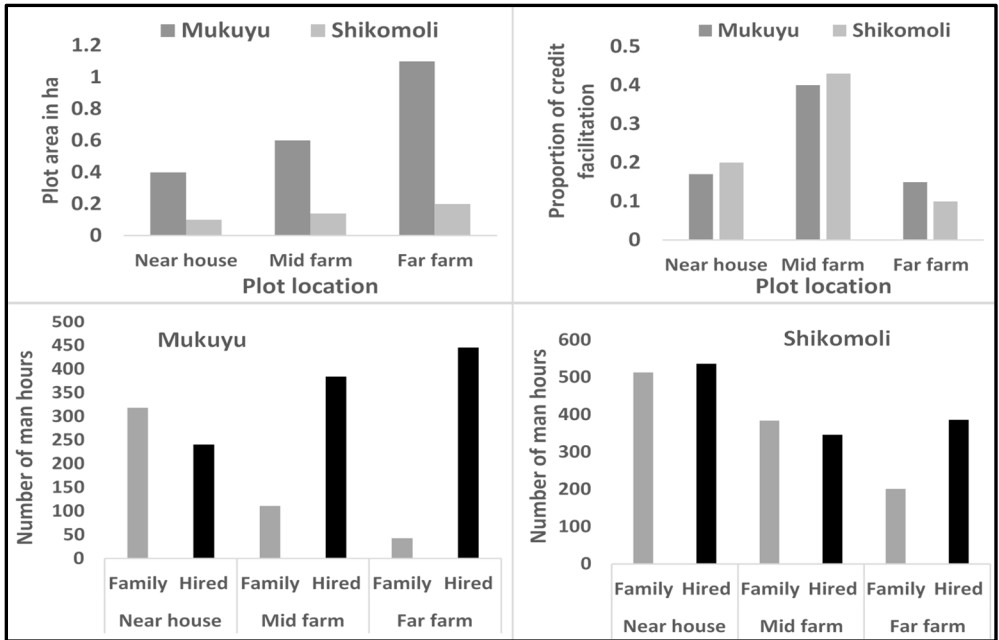

**Figure 11.** Socio-economic factors: land allocation, gender, credit services and labor use in maize production.

## 4. Discussion

### 4.1. Yield Gap Patterns at Different Spatial Arrangements

The highly diverse smallholder farming conditions implies that yield gap mapping needs to acknowledge existence of variability on smallholder farms [37]. That is, mapping should indicate yield gaps at different levels (high, low and average). The results demonstrated the potential use of spatial arrangements on smallholder farms to show yield gap variability and field-specific utilization of factors of production—management, socio-economic and biophysical factors. Patterns of high, median and low yield gaps which were mapped on spatial arrangements closer to the homestead was an indication of heterogeneous yield patterns. The findings coincide with [38], who have shown spatial yield variability patterns on smallholder farms, attributed to management, soil and climatic factors. As the distance increased from the homestead, the high and low yield gap patterns stretched towards nearly homogenous maize yield gap patterns, indicating better performing fields. The results are congruent with [39], who has shown better management and performance with increasing distance from the homestead.

Heterogeneous patterns at the near house spatial arrangements which were also shown by high variance in yield gaps indicated unequal use of management and socio-economic factors. Some sections of near house spatial arrangements could have received better management practices such as high organic fertilizer application, and had high plant density (Figure 9). This contributed to high maize yields and low yield gaps. Other fields or sections of the near house spatial arrangements had delayed management practices such as late land preparation, untimely weed control, utilization of

short maize variety, less land allocation and low proportion of credit facility use (Figures 9 and 10). This could have contributed to high yield gaps compared to mid and far farm spatial arrangements. The inconsistence in utilization of field level management practices and socio-economic factors thus contributed to heterogeneous patterns of low and high yield gaps. Patterns of high yield gap could also have resulted from sections of the near house spatial arrangements having high phyto-diversity which reduced plant population resulting in low yield [40,41].

Nearly homogenous patterns of yield gaps on the mid and far farm spatial arrangements could have resulted from consistent utilization of management and socio-economic factors such as high weed frequency, inorganic fertilizer use, high maize density, early land preparation, long and medium maize variety use (Figure 9). Smallholder farmers operate under resource constraints and tend to minimize management and socio-economic resources to achieve large coverage [42]. Therefore, the nearly homogenous yield gap patterns could also have resulted from minimal resource use spread over the entire field due to the large land size allocated to the mid and far farm spatial arrangements (Figure 9). All sections within the fields received almost nearly equal treatment.

Studies have shown that farmers manage certain fields within farming systems according to certain perceived benefits [43]. The consistent utilization of management practices and socio-economic factors at mid and far farm spatial arrangements indicated preferential treatment of these fields over the near house spatial arrangements which could be due to certain perceived benefits that need to be investigated. This was corroborated by the positive correlation of biophysical factors phosphorus (P), nitrogen (N) and chlorophyll (SPAD1 and SPAD3 values) (Table A1, see Appendix A), and negative correlation of weed pressure and weed height with increasing distance from the homestead (PLOTDist) (Table A2, see Appendix A).

## 4.2. The Production Opportunities for the Different Spatial Arrangements to Enhance Maize Yields

Managing heterogeneity in management, socio-economic and biophysical factors on smallholder farms has been identified as one avenue through which to increase and sustain food production [42]. Identifying field zones with similar yield gap patterns and management can aid in designing soil and crop measures that could be applied uniformly within a certain agro-ecology. In this study, focal statistics was applied to generate yield gap maps based on different spatial arrangements found on smallholder farms. The spatial arrangements illustrated different yield gap patterns: heterogeneous and nearly homogenous. The survey findings revealed nearly uniform utilization of socio-economic and management factors at spatial arrangements further away from the homestead compared to ones closer to the homestead. Increasing utilization of socio-economic and management factors of spatial arrangements further away from the homestead could therefore lead to high and consistent yields and low yield gaps within the farm.

The different spatial arrangements also depicted production opportunities regarding management, socio-economic and biophysical factors that could be utilized to improve maize yields. The high proportion of organic fertilizer used at the near house plots indicated a high nutrient supply and water retention [44]. This can be utilized to increase maize production by improving timely execution of agronomic activities such as land preparation, weed control and use of long duration maize varieties. The high proportion of inorganic fertilizer use at the mid and far farm plots is an indication of increased nutrient supply which can efficiently be utilized by timely management of weeds to achieve high yield. Increasing plant density of the plots at mid and far sections of the farm which have large land sizes, by adopting an optimal plant spacing can also help maximize land use and resources.

The low-lying terrain of the far farm fields in Mukuyu was an indication of increased nutrient accumulation washed down from the mid and near house plots when there is a heavy downpour [45]. This benefit can be utilized to enhance yields by increasing plant density of the far farm fields. Family labor provides a supervisory role to ensure resources such as fertilizer, seeds and time are used efficiently to increase productivity [46]. In Mukuyu, the reliance on utilization of hired labor at the mid and far farm fields could indicate reduced resource use efficiency with subsequent effect on soil

fertility and productivity. There is a need to increase utilization of family labor for the mid farm plots to maximize resource utilization. Scheduling farm activities such as planting and weed control to coincide with availability of family members to provide a supervisory role will help improve labor utilization and resource use efficiency.

## 5. Conclusions

The study demonstrated the use of spatial arrangements found on smallholder farms as a unique approach to identify patterns of yield gap variability and survey data to reveal field-specific utilization of management, socio-economic and biophysical factors as a scope for enhancing site-specific land management. The findings demonstrated different patterns of low, median and high maize yield gaps. When yield gaps were mapped on spatial arrangements closer to the homestead, highly heterogeneous patterns of low, median and high yield gaps were realized. As distances increased from the homestead, nearly homogenous patterns of median to high yield gaps were found. Survey investigation using management and socio-economic factors further explains the occurrence of the yield gap patterns. Delineating management zones based on yield gap patterns at the different spatial arrangements on smallholder farms could contribute to site-specific land management and enhance yields. The findings also revealed that smallholder farmers preferentially manage spatial arrangements further from the homestead regarding application of socio-economic and management factors than near house fields. The challenge now remains upon how to increase the consistency in utilization and replication of these factors in spatial arrangements further from and closer to the homestead in order to enhance yields. Investigating the value smallholder farmers attach for each spatial arrangement can further enhance the spatial understanding of yield gap variation on smallholder farms.

**Author Contributions:** Conceptualization, M.S., H.O., G.D. M.H; methodology, M.S., H.O., G.D. and B.-O.N.; software, M.S. and F.A.B.M.; validation; H.O., B.-O.N.; formal analysis, M.S. and H.O.; investigation, G.D.; data preparation; M.S., H.O., B.-O.N., F.A.B.M. writing—original draft preparation, M.S.; writing—review and editing, M.S., H.O., G.D., O.M.C., O.-K.W.; visualization, M.S.; supervision, H.O., O.M.C., O.-K.W., G.D.; project administration, G.D., O.-K.W., M.H.

**Funding:** This research was funded by Swedish Research Councils Formas [220–2014–646] and VR [04440].

**Acknowledgments:** The authors gratefully to smallholder farmers in Mukuyu and Shikomoli who willingly donated portions of their farms for field activities. Similarly, appreciation is due to the field guides and local authority in both study sites for providing guidance during the field activities.

**Conflicts of Interest:** The authors declare no conflict of interest. The funders had no role in the design of the study; in the collection, analyses, or interpretation of data; in the writing of the manuscript, or in the decision to publish the results.

# Appendix A

**Table A1.** Correlation Matrix for Mukuyu.

|  | Intercept | P | N | WC1 | WH1 | MDD1 | MH3 | SPAD3 | SPAD1 | MH1 | WC3 | PLOTDist | QTIng | TTLS | MDD3 | B |
|---|---|---|---|---|---|---|---|---|---|---|---|---|---|---|---|---|
|  | 1 | | | | | | | | | | | | | | | |
| P | −0.052 | 1 | | | | | | | | | | | | | | |
| N | −0.158 | −0.078 | 1 | | | | | | | | | | | | | |
| WC1 | 0.262 | −0.136 | −0.112 | 1 | | | | | | | | | | | | |
| WH1 | 0.172 | −0.037 | −0.025 | −0.27 | 1 | | | | | | | | | | | |
| MDD1 | −0.517 | −0.021 | −0.242 | −0.114 | 0.176 | 1 | | | | | | | | | | |
| MH3 | −0.136 | −0.02 | −0.328 | 0.044 | −0.043 | 0.129 | 1 | | | | | | | | | |
| SPAD3 | −0.487 | −0.246 | 0.014 | −0.391 | 0.098 | 0.195 | 0.181 | 1 | | | | | | | | |
| SPAD1 | −0.49 | −0.105 | 0.269 | −0.016 | 0.245 | −0.001 | −0.486 | 0.158 | 1 | | | | | | | |
| MH1 | −0.365 | 0.058 | −0.077 | −0.117 | −0.725 | −0.171 | −0.058 | 0.259 | −0.26 | 1 | | | | | | |
| WC3 | 0.224 | 0.049 | −0.015 | −0.169 | 0.04 | 0.18 | 0.178 | −0.043 | −0.076 | −0.049 | 1 | | | | | |
| PLOTDist | −0.336 | 0.256 | 0.238 | −0.108 | −0.141 | 0.269 | 0.241 | 0.069 | 0.027 | 0.015 | 0.121 | 1 | | | | |
| QTIng | −0.108 | 0.158 | −0.142 | −0.074 | −0.117 | 0.216 | 0.064 | −0.227 | −0.007 | 0.185 | 0.172 | 0.192 | 1 | | | |
| TTLS | −0.182 | −0.063 | −0.121 | −0.121 | −0.128 | 0.026 | −0.01 | −0.163 | −0.234 | 0.106 | −0.169 | −0.349 | −0.017 | 1 | | |
| MDD3 | 0.059 | −0.053 | 0.034 | 0.153 | 0.089 | −0.681 | −0.275 | 0.061 | 0.092 | −0.053 | −0.13 | −0.033 | −0.08 | −0.26 | 1 | |
| B | −0.055 | −0.052 | −0.209 | −0.02 | −0.062 | −0.075 | 0.044 | 0.057 | −0.048 | 0.082 | 0.021 | 0.121 | 0.119 | 0.25 | −0.031 | 1 |

Legend: P—Phosphorus, N—Nitrogen, WC1—Weed cover in stage 1 of maize development, WH1—Weed height in stage 1, MDD1—Maize density in stage 1, MH3—Maize height in stage 3, Soil Plant Analysis Development (SPAD3)—Chlorophyll content in stage 3, SPAD 1—Chlorophyll content in stage 3, MH1—Maize height in stage 1, WC3—Weed height in stage 3, PLOTDist—Distance of the spatial arrangement form the homestead, TTLS—Total land size, MDD3—Maize density in stage 3, B—Boron.

**Table A2.** Correlation Matrix for Shikomoli.

| | Intercept | P | N | WC1 | WH1 | MDD1 | MH1 | SPAD3 | SPAD1 | MH1 | WC3 | PLOTDs | QtIng | TTLS | MDD3 | B |
|---|---|---|---|---|---|---|---|---|---|---|---|---|---|---|---|---|
| Intercept | 1 | | | | | | | | | | | | | | | |
| P | −0.08 | 1 | | | | | | | | | | | | | | |
| N | −0.358 | −0.357 | 1 | | | | | | | | | | | | | |
| WC1 | 0.164 | 0.27 | −0.055 | 1 | | | | | | | | | | | | |
| WH1 | 0.045 | 0.054 | −0.089 | −0.155 | 1 | | | | | | | | | | | |
| MDD1 | −0.279 | 0.12 | −0.066 | −0.232 | −0.022 | 1 | | | | | | | | | | |
| MH1 | 0.157 | −0.064 | −0.144 | −0.042 | −0.054 | −0.171 | 1 | | | | | | | | | |
| SPAD3 | 0.034 | 0.007 | −0.067 | −0.154 | 0.104 | 0.105 | −0.014 | 1 | | | | | | | | |
| SPAD1 | −0.699 | 0.232 | 0.069 | 0.117 | −0.042 | 0.299 | −0.225 | −0.143 | 1 | | | | | | | |
| MH1 | −0.186 | −0.138 | 0.111 | −0.088 | −0.415 | −0.006 | −0.401 | −0.145 | −0.079 | 1 | | | | | | |
| WC3 | −0.178 | −0.069 | 0.074 | −0.285 | −0.087 | −0.053 | 0.125 | −0.211 | 0.082 | 0.136 | 1 | | | | | |
| PLOTDs | 0.001 | −0.065 | 0.048 | −0.122 | −0.028 | −0.025 | −0.048 | −0.195 | −0.038 | 0.138 | −0.091 | 1 | | | | |
| QtIng | −0.031 | −0.047 | 0.031 | 0.055 | 0.043 | −0.147 | 0.173 | 0.119 | −0.119 | −0.118 | −0.127 | −0.035 | 1 | | | |
| TTLS | −0.363 | 0.257 | −0.069 | 0.094 | 0.063 | −0.021 | −0.144 | −0.256 | 0.319 | 0.089 | 0.176 | −0.188 | 0.159 | 1 | | |
| MDD3 | −0.114 | 0.032 | −0.02 | −0.04 | 0.06 | −0.365 | −0.198 | −0.239 | −0.192 | 0.059 | 0.098 | −0.056 | 0.038 | 0.179 | 1 | |
| B | −0.006 | 0.138 | −0.164 | 0.135 | 0.02 | 0.006 | −0.129 | −0.036 | 0.027 | 0.014 | 0.113 | −0.049 | −0.022 | 0.175 | 0.23 | 1 |

Legend: P—Phosphorus, N—Nitrogen, WC1—Weed cover in stage 1 of maize development, WH1—Weed height in stage 1, MDD1—Maize density in stage 1, MH3—Maize height in stage 3, Soil Plant Analysis Development (SPAD3)—Chlorophyll content in stage 3, SPAD 1—Chlorophyll content in stage 3, MH1—Maize height in stage 1, WC3—Weed height in stage 3, PLOTDist—Distance of the spatial arrangement form the homestead, TTLS—Total land size, MDD3—Maize density in stage 3, B—Boron.

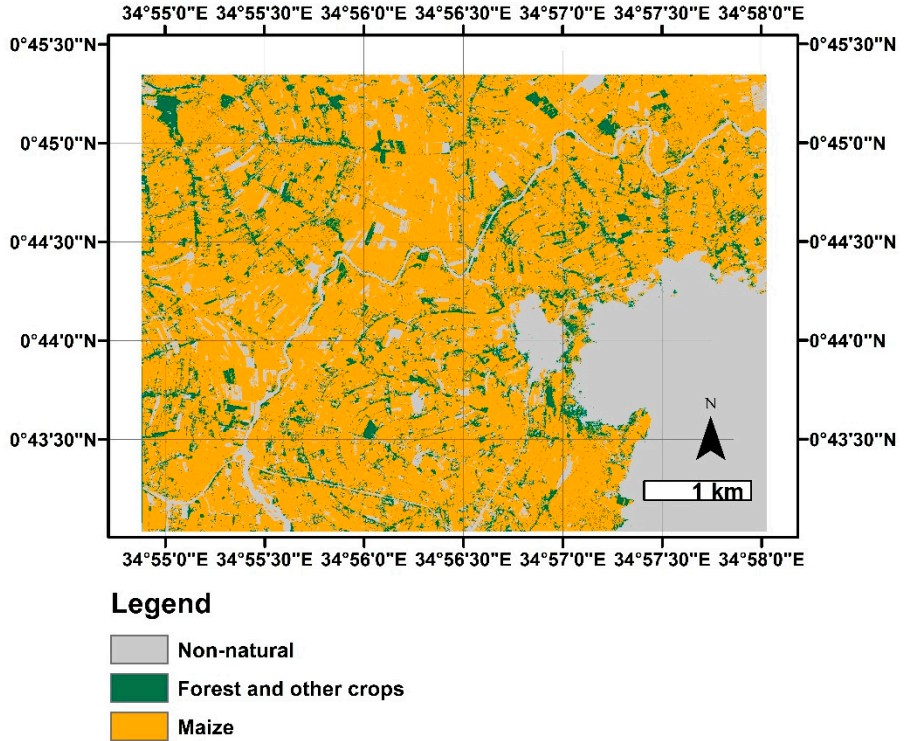

**Figure A1.** Classification of non-natural, forest and other crops, and maize land in Mukuyu.

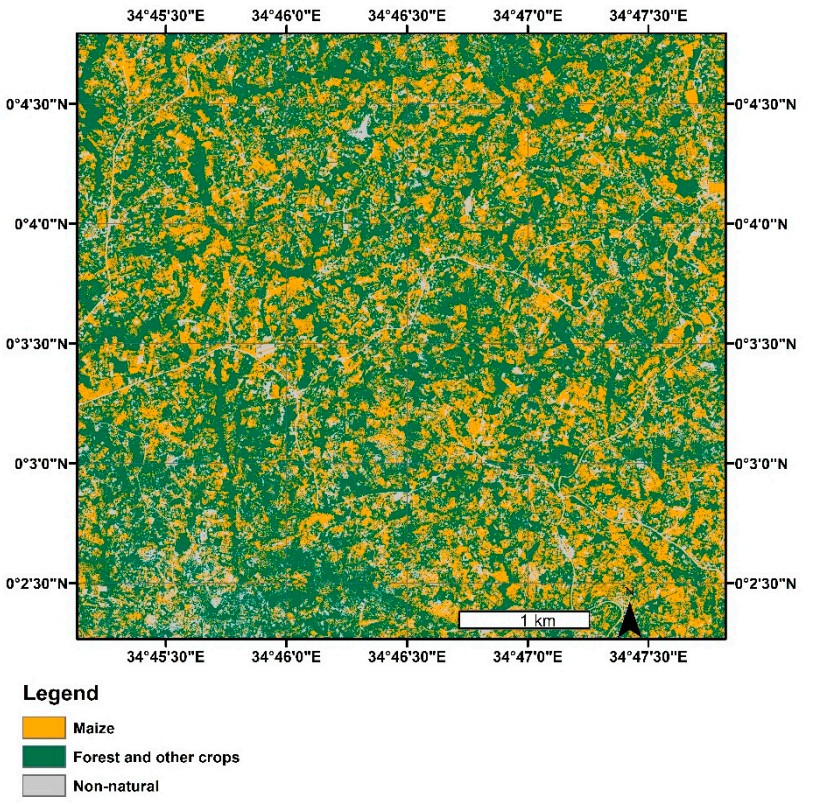

**Figure A2.** Classification of non-natural, forest and other crops, and maize in Shikomoli.

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
