# Peer review of "Micro-Spatial Analysis of Maize Yield Gap Variability and Production Factors on Smallholder Farms"

_agriculture, doi:10.3390/agriculture9100219_

Round 1

Reviewer 1 Report

The manuscript aims at understanding field variability based on yield
gap mapping patterns in order to enhance resource use efficiency on
smallholder farms. Two test sites with different agro-ecological
potential, different population density and different market
accessibility have been investigated using optical data and in-situ
measurements. The authors identifies three spatial arrangements (near house, mid and far farm) to which they analysis yield gap mapping
patterns in terms of fertilizer application per ha, organic
fertilizer use, land preparation and erosion status. There is found
that smallholder farmers preferentially manage the application of
socio-economic and management factors on spatial arrangements further
the homestead compared to fields closer the homestead. In conclusion,
investigating the value smallholder farmers attach for each spatial
arrangement can enhance the spatial understanding of yield gap
variation on smallholder farms.

L - 213 Landsat 8 instead of Land sat 8 L- 220 Landsat 8
Please details the methodology and offers information about in-situ and ancillary data (when have been collected) in order to reproduce them.

Author Response

L - 213 Landsat 8 instead of Land sat 8 L- 220 Landsat 8

Response: The word has been revised L-236 and 243

Please details the methodology and offers information about in-situ and ancillary data (when have been collected) in order to reproduce them.

Response: A detailed methodology has been provided to describe how socio-economic, biophysical and management factors were collected. L-199-L-220

Reviewer 2 Report

Micro-Spatial Analysis of Maize Yield Gap Variability and Production Factors on Smallholder Farms

The manuscript focuses on yield gaps forsmallholder fields based on identified spatial arrangements differentiated by distance from the smallholder homestead for understanding field specific utilization of production factors and thus help on enhancing their resource use efficiency. The study took place in two villages (Mukuyu and Shikomoli) with high and low agroecology regarding soil fertility in Western Kenya, while the spatial arrangements of small holder farms were located at 40m, 80m, 150m and 300m distance from the homestead. Socio-economic, management and biophysical factors were assessed and maize yields were estimated at each spatial arrangement. The authors found that there was uniform utilization of socio-economic and management factors at spatial arrangements further the homestead compared to ones closer the homestead. Finally, they concluded that further investigation is needed on the value smallholder farmers attach for each spatial arrangement in order to enhance the spatial understanding of yield gap variation on smallholder farms.

Comments

L106. The authors need to use past tense since the study has already been conducted.

L110-116. The authors need to use past tense since the study has already been conducted.

L134. The degrees need to be written correct for not confusing the readers of the manuscript.

L196-197. The authors need to use “was” instead of “were”.

L198. What was the temperature used (750 oC or 75 oC)?

L213-214. The authors need to rephrase this sentence.

L233-234. The authors need to provide a proper reference for the methodology that they used in R.

L249. The authors need to properly write the software that they used e.g. ArcGIS 10.5 (ESRI Inc., Redlands, USA).

L362-363. The authors need to rephrase this sentence.

L363. The authors need to refer in detail to the use of SPAD measurements in the materials and methods section.

L386-397. The authors need to use past tense.

L426-455. The authors need to use past tense.

L456-466. This section needs to be integrated in the Conclusions section.

General Comment

The authors used a sound methodology to identify patterns of yield gap variability based on different spatial arrangements of the small holder farms when compared from the homestead distance. In this way, their study will help on revealing field specific utilization of management, socio-economic and biophysical factors as scope in order to enhance site specific land management.

Author Response

Reviewer two feedback

L106. The authors need to use past tense since the study has already been conducted.

Response: The authors have revised to past tense L106

L110-116. The authors need to use past tense since the study has already been conducted.

Response: The authors have revised to past tense L110-116

L134. The degrees need to be written correct for not confusing the readers of the manuscript.

Response: The degrees have been revised L157

L196-197. The authors need to use “was” instead of “were”.

Response: The authors have revised the word L220

L198. What was the temperature used (750 oC or 75 oC)?

Response: The temperature has been changed to 75 oC. L221

L213-214. The authors need to rephrase this sentence.

Response: The authors have rephrased the sentence L237-L239

L233-234. The authors need to provide a proper reference for the methodology that they used in R.

Response: The authors have added a reference for the methodology that was used L257

L249. The authors need to properly write the software that they used e.g. ArcGIS 10.5 (ESRI Inc., Redlands, USA).

Response: The authors have added a reference for the methodology that was used L274

L362-363. The authors need to rephrase this sentence.

Response: The authors have rephrased the sentence L398-L399

L363. The authors need to refer in detail to the use of SPAD measurements in the materials and methods section.

Response: The authors have provided details to describe SPAD measurements and other socio-economic, management and biophysical factors L199-L218

L386-397. The authors need to use past tense.

Response: The authors have used past tense L421-L433

L426-455. The authors need to use past tense.

Response: The authors have used past tense L466-472

L456-466. This section needs to be integrated in the Conclusions section.

Response: The authors integrated the section in one of the headings in the discussion section L462-472 and partially in the discussion section L494-L509

This manuscript is a resubmission of an earlier submission. The following is a list of the peer review reports and author responses from that submission.